# On Computational Hardness of Multidimensional Subtraction Games †

**Vladimir Gurvich * and Mikhail Vyalyi **

Faculty of Computer Science/Big Data and Information Retrieval School, HSE University,
109028 Moscow, Russia; vyalyi@gmail.com
* Correspondence: vladimir.gurvich@gmail.com
† This paper is an extended version of our paper published in CSR 2020: 15th International Computer Science Symposium in Russia.

**Abstract:** We study the algorithmic complexity of solving subtraction games in a fixed dimension with a finite difference set. We prove that there exists a game in this class such that solving the game is **EXP**-complete and requires time $2^{\Omega(n)}$, where $n$ is the input size. This bound is optimal up to a polynomial speed-up. The results are based on a construction introduced by Larsson and Wästlund. It relates subtraction games and cellular automata.

**Keywords:** subtraction games; cellular automata; computational hardness

## 1. Introduction

The algorithmic complexity of solving combinatorial games is an important part of algorithmic game theory. Some famous games can be solved efficiently. For example, the ancient game of nim was solved by Bouton in [1].

A position $x$ of nim is determined by $n$ heaps of pebbles. It can be considered to be a $n$-dimensional integer vector. By one move it is allowed to reduce (strictly) exactly one heap. Two players move alternately. One who is out of moves loses. In fact, in his paper Bouton obtained an explicit formula for the Sprague-Grundy (SG) function of the disjunctive compound (or, for brevity, the sum) of impartial games. Informally, the SG function is a homomorphism of an impartial game to nim with one heap. In particular, $SG(x) > 0$ and $SG(x) = 0$ indicate that the player who moves at the position $x$ and, respectively, the opponent has a winning strategy; such positions are called N- and P-positions. See Section 2.1 and [2–4] for more details.

It is important for our purposes that the formula for the SG function of nim can be computed efficiently even if the sizes of the heaps are given in binary notation.

Similar efficient solutions were found for several versions and/or generalizations of nim: Wythoff's nim [5,6], Fraenkel's nim [6,7], nim$(a, b)$ game [8], Moore's $(n, k)$-nim with $k = n - 1$ [9–11], and the exact $(n, k)$-nim with $2k \geq n$ [11,12]. In all these versions it is allowed to reduce by one move several heaps, not necessarily only one. Specifying the rules of choosing heaps and the numbers of pebbles that can be taken, one gets different versions of nim. For example, in the exact (resp., Moore) $(n, k)$-nim a player by one move reduces exactly (resp., at most) $k$ from $n$ heaps, strictly but otherwise arbitrarily, where $n$ and $k$ are fixed integer parameters such that $0 < k < n$.

Another way to change the rules is to restrict moves to *slow* ones. A move is called *slow* if at most one pebble is taken from each heap. The Moore and exact *slow* $(n, k)$-nim were introduced in [13,14]. In the study of the exact slow $(n, k)$-nim explicit and efficiently computable formulas for the SG function were obtained for $(n, k) = (3, 2)$ [13] and for $(n, k) = (4, 2)$ [14]. However, for larger values of $k$, no explicit formula for the SG function of the exact or Moore slow $(n, k)$-nim is known. Moreover, numerical experiments show that even P-positions look rather chaotic.

Is it possible that there is no efficient algorithm solving these variants of nim? Right now, we have no answer to this question and even no clue.

As for hardness results, we could mention numerous examples of **PSPACE**-complete combinatorial games (see, e.g., [15,16]) and also **NP**-hardness of hypergraph nim for some classes of hypergraphs [17,18].

The game *hypergraph nim*, $\text{NIM}_{\mathcal{H}}$, is specified by a hypergraph $\mathcal{H} \subseteq 2^{[n]} \setminus \{\varnothing\}$ on the ground set $[n] = \{1, \ldots, n\}$. It is played as follows. By one move a player chooses an edge $H \in \mathcal{H}$ and strictly reduces all heaps of $H$. The games of standard (not slow) exact and Moore's nim considered above provide examples of the hypergraph nim. The *height* $h(x) = h_{\mathcal{H}}(x)$ of a position $x = (x_1, \ldots, x_n)$ of $\text{NIM}_{\mathcal{H}}$ is the maximum number of successive moves that players can make beginning in $x$. (Clearly, they can restrict themselves by their slow moves.) A hypergraph $\mathcal{H}$ is called *intersecting* if $H' \cap H'' \neq \varnothing$ for any two edges $H', H'' \in \mathcal{H}$. It was proved in [17,18] that for any intersecting hypergraph $\mathcal{H}$, its height and SG function are equal. In [17] it was proved that computing $h_{\mathcal{H}}(x)$ is **NP**-hard already for the intersecting hypergraphs with edges of size at most 4. Thus, **NP**-hardness of computing the SG function for such family of hypergraphs follows from these two statements.

This example is a typical hardness result in this area. Hardness is established for growing 'dimension' of games (in this case, 'dimension' is the number of heaps). For the case of a fixed number of heaps Larsson and Wästlund [19] obtained an important result. They studied a wider class of games, so-called *vector subtraction games* (subtraction games for brevity). These games were introduced by Golomb [20], they also were studied under the name of *invariant games* [21]. The subtraction games include all versions of nim mentioned above. In these games, the positions are $d$-dimensional vectors with nonnegative integer coordinates. A move in a position is a subtraction of a 'permitted' vector from the vector determining this position. The set of 'permitted' vectors is called the *difference set*. Thus the game is completely specified by the difference set. Larsson and Wästlund considered subtraction games of finite dimension with a finite difference set (FDG for brevity: fixed dimension and fixed difference set).

It is easy to see that the P-positions of a 1-dimensional FDG form an eventually periodic set [22] (actually, the values of the SG function are also eventually periodic in this case). It gives an efficient algorithm of solving such games. Yet, the FDG of higher dimensions may behave in a highly complicated way. More exactly, Larsson and Wästlund proved in [19] that in some fixed dimension the equivalence problem for FDG is undecidable.

However, this remarkable result does not answer the main question: whether efficient algorithms solving FDG exist. For example, there are polynomial algorithms solving the membership problem for context-free languages but the equivalence problem for them is undecidable [23].

In this paper, we extend arguments of Larsson and Wästlund and prove the existence of a FDG such that solving the game is **EXP**-complete and requires time $2^{\Omega(n)}$, where $n$ is the input size. Furthermore, this bound is optimal up to a polynomial speed-up.

The rest of the paper is organized as follows. In Section 2 we introduce the concepts that we will need and outline our contribution. The proof of the main theorem is outlined in Section 3. The following sections contain some more detailed exposition of main steps of the proof. In Section 4 we describe a relation between binary cellular automata and subtraction games. In Section 5 we describe a simulation of a Turing machine by a binary cellular automaton. In Section 6 we present a way to launch a Turing machine on all inputs simultaneously. Section 7 contains the main proof itself. Finally, in Section 8 we discuss possible lines of future research in this direction.

## 2. Concepts and Results

### 2.1. Impartial Games

Positions and moves of an *impartial* game form a directed acyclic graph (DAG). It implies that positions in a play cannot be repeated. Such DAG may be infinite, but the set of positions reachable (by one or several moves) from any given position is finite. A choice

of an initial position specifies an instance of a game. Two players move alternately. One who must move but is out of moves loses.

Positions of an impartial game are divided in N-*positions* and P-*positions* (from words Next and Previous). A player who moves at an N-position has a winning strategy. A player who moves at a P-position has not a winning strategy (therefore the opponent has a winning strategy). In graph theory, the set of P-positions of a DAG is called its *kernel*; it can be found in time linear in the size of the DAG [24]. It gives an efficient algorithm solving an impartial game represented by the list of positions and moves. However, many games admit succinct description. The size of DAG may be exponential in size of the succinct description. It makes the kernel construction algorithm exponential.

The Sprague-Grundy (SG) function refines the concept of N- and P-positions. For a set $S$ of nonnegative integers the *minimum excluded value* of $S$ is defined as the smallest nonnegative integer that is not in $S$ and denoted by $\mathrm{mex}(S)$. In particular, $\mathrm{mex}(\varnothing) = 0$. The SG value of a position $x$ is defined recursively as

$$SG(x) = \mathrm{mex}\{SG(y) : (x,y) \in E(G)\}, \tag{1}$$

where $G$ is the corresponding DAG. A position of SG value $t$ is called a $t$-position. Then, P-positions are exactly 0-positions, while N-positions have positive SG values.

Taking in mind the relation with the Sprague-Grundy function, we assign to P- and N-positions the (Boolean) values 0 and 1, respectively. The basic relation between values of positions is

$$p(v) = [p(v_1), \ldots, p(v_k)] = \neg \bigwedge_{i=1}^{k} p(v_i) = \bigvee_{i=1}^{k} \neg p(v_i), \tag{2}$$

where all the possible moves from the position $v$ are to the positions $v_1, \ldots, v_k$. If $v$ is a sink then $p(v) = 0$. We will use notation $[\ldots]$ introduced in [19] for Boolean functions in Equation (2).

### 2.2. Subtraction Games and Modular Games

Now we introduce the class FDG of subtraction games. Subtraction games generalize naturally all versions of nim mentioned above. Please note that a position in a version of nim with $d$ heaps is specified by a $d$-dimensional vector $x = (x_1, \ldots, x_d)$ with nonnegative integer coordinates which are just the numbers of pebbles in a heap. A move in the game decreases some coordinates of this vector. Thus, possible moves are specified by a set $\mathcal{D}(x) \subseteq \mathbb{N}^d$ of $d$-dimensional vectors with nonnegative integer coordinates (the *difference set*). A move from $x$ to $y$ is possible if $x - y \in \mathcal{D}(x)$.

**Example 1.** *Suppose that $x = (x_1, \ldots, x_n)$ is a position in the exact slow $(n,k)$-nim introduced in Section 1. Any legal move in this position should decrease exactly $k$ values of the coordinates by 1. Therefore $\mathcal{D}(x)$ is the set of $d$-dimensional $(0,1)$-vectors with exactly $k$ ones, for any $x$.*

A subtraction game is defined by similar rules. There are two important differences. In a subtraction game the difference set $\mathcal{D}$ is the same for all positions. In a general subtraction game $\mathcal{D}$ may contain integer vectors with negative coordinates. It terms of heaps and pebbles it means that a player is allowed to add pebbles to heaps. To avoid infinite plays, we require that the total number of pebbles should be strictly reduced by each move. It is equivalent to the following condition on coordinates of difference vectors:

$$\sum_{i=1}^{d} a_i > 0 \tag{3}$$

for any difference vector $a \in D$.

Finally, a game from the class FDG is defined by a *finite* difference set $\mathcal{D}$ of $d$-dimensional vectors satisfying (3). In particular, the exact slow $(n,k)$-nim belongs to FDG.

**Example 2.** *Let* $\mathcal{D} = \{(2, -1), (-1, 2)\}$. *Then possible moves from position* $(3, 3)$ *are to positions* $(1, 4)$ *and* $(4, 1)$. *It is an easy exercise to compute the value of the position:* $p(3, 3) = 0$, *i.e., it is a P-position.*

Due to (3), any play of a FDG starting at a position $(x_1, \ldots, x_d)$ includes only positions with smaller sums of the coordinates. Therefore, computing the value of $(x_1, \ldots, x_d)$ by the basic recurrence relation (2) requires the values at most

$$\binom{M + d}{d} = O(M^d)$$

positions, where $M = \sum_i x_i$. Therefore, due to the kernel construction algorithm mentioned above, solving FDG belongs to the class **EXP** if the $x_i$ are presented in binary notation.

This exponential bound on the complexity of solving FDG holds in the case when the difference set is a part of the input. It is easy to show that in this case solving FDG is **PSPACE**-hard. For this purpose, we reduce solving the game called NODE KAYLES to solving a FDG. Recall the rules of NODE KAYLES. It is played on a graph $G$. At each move a player puts a pebble on an unoccupied vertex of $G$ that is non-adjacent to any occupied vertex. As usual, the player unable to make a move loses.

**Proposition 1.** *Solving NODE KAYLES is polynomially reducible to solving FDG.*

**Proof.** Let $G = (V, E)$ be a graph of NODE KAYLES. Construct a $|E|$-dimensional subtraction game with the difference set $A_G$ indexed by the vertices of $G$: $D = \{a^{(v)} : v \in V\}$, where

$$a_e^{(v)} = \begin{cases} 1, & \text{the vertex } v \text{ is incident to the edge } e, \\ 0, & \text{otherwise.} \end{cases}$$

We assume in the definition that the coordinates are indexed by the edges of the graph $G$.

Take the position $\mathbb{1}$ with all coordinates equal 1. We are going to prove that this position is a P-position of the FDG $A_G$ iff the graph $G$ is a P-position of NODE KAYLES.

Indeed, after subtracting a vector $a^{(v)}$, coordinates indexed by the edges incident to $v$ are zero. It means that after this move it is impossible to subtract the vector $a^{(v)}$ and any vector $a^{(u)}$ such that $(u, v) \in E$.

On the other hand, if the current position is

$$\mathbb{1} - \sum_{v \in X} a^{(v)}$$

and there are no edges between a vertex $u$ and the vertices of the set $X$, then the subtraction of the vector $a^{(u)}$ is a legal move at this position.

Thus, the subtraction game starting from the position $\mathbb{1}$ is isomorphic to the game NODE KAYLES on the graph $G$.  □

Since solving NODE KAYLES is **PSPACE**-complete [15], we get **PSPACE**-hardness of solving FDG as an immediate corollary of Proposition 1.

In the rest of the paper, we consider a more restricted case. Specifically, we assume that the difference set $\mathcal{D}$ is fixed and an instance of a computational problem is a vector $(x_1, \ldots, x_d)$ specifying a position, and the coordinates of the vector are presented in binary notation. The problem is to decide whether $(x_1, \ldots, x_d)$ is a P-position. In other words, we are going to determine algorithmic complexity of the language $\mathcal{P}(\mathcal{D})$ consisting of the binary representations of all P-positions $(x_1, \ldots, x_d)$ of the FDG with the difference set $\mathcal{D}$.

Our main result is unconditional hardness of this problem.

**Theorem 1.** *There exist a constant d and a finite set* $\mathcal{D} \subset \mathbb{N}^d$ *such that the language* $\mathcal{P}(\mathcal{D})$ *is* **EXP**-*complete and* $\mathcal{P}(\mathcal{D}) \notin$ **DTIME**$(2^{\Omega(n)})$, *where n is the input size.*

As an immediate corollary of Theorem 1 we see that for some languages $\mathcal{P}(\mathcal{D})$ the kernel construction algorithm is optimal up to polynomial speed-up.

In the proofs we need a more general class of games introduced in [19]. They are called *k-modular FDG*. A *k*-modular *d*-dimensional FDG is determined by $k$ finite sets $D_0, \ldots, D_{k-1}$ of vectors from $\mathbb{Z}^d$. Each vector from each $D_i$ should satisfy the condition (3). The rules are similar to those of FDG, but the set of possible moves at a position $x$ is $D_r$, where $r$ is the residue of $\sum_i x_i$ modulo $k$.

**Example 3.** *Let $D_0 = \{(1,0), (0,1)\}$, $D_1 = \{(2,-1), (-1,2)\}$. Then the possible moves from position $(3,3)$ in 2-modular game $D_0, D_1$ are to positions $(3,2)$ and $(2,3)$ (since $3+3=6$ is even). Possible moves from $(2,3)$ are to $(0,4)$ and $(3,1)$ (since $2+3$ is odd).*

*2.3. Turing Machines and Cellular Automata*

Turing machines are a well-known universal model of computation. For the formal definition we refer to Sipser's book [25].

Although the cellular automata are also well-known, for the reader's convenience we provide the definition. Informally, a cellular automaton operates on a tape consisting of *cells*. The tape is infinite in both directions. Each cell carries a symbol from some fixed set which is called the *alphabet*. At each step all the symbols in cells are changed simultaneously. The change is governed by the same rule for all cells. This rule specifies a new symbol in a cell depending on a content of a fixed neighborhood of the cell. Formally, a cellular automaton (CA) $C$ is a pair $(A, \delta)$, where $A$ is an alphabet, and $\delta \colon A^{2r+1} \to A$ is the *transition function*. The number $r$ is called *the size of a neighborhood*. A *configuration* of $C$ is a function $c \colon \mathbb{Z} \to A$.

An operation of a CA starting at a configuration $c_0$ generates the infinite sequence of configurations. The sequence is defined by the recurrence

$$c_{t+1}(u) = \delta\big(c_t(u-r), c_t(u-r+1), \ldots, c_t(u), \ldots, c_t(u+r-1), c_t(u+r)\big).$$

Please note that the changes are local: the content of a cell depends only on the content of $2r + 1$ neighbor cells. It is easy to see that these formal definitions fit the informal description of an operation of a CA.

We assume that there exists a blank symbol $\Lambda$ in the alphabet and the transition function satisfies the condition $\delta(\Lambda, \ldots, \Lambda) = \Lambda$ ("nothing will come of nothing"). If a CA with the blank symbol starts from a configuration containing only a finite number of non-blank symbols, then its operation generates the sequence of configurations with the same property.

A 2CA (a binary CA) is a CA with the binary alphabet $\{0, 1\}$. We assume that 1 is the blank symbol in 2CAs. This non-standard assignment is convenient due to connections with games.

**Example 4.** *Let $C = (\{0,1\}, \delta(x_{-1}, x_0, x_1) = x_{-1} \oplus x_1 \oplus 1)$. It is a 2CA with $r = 1$ and the blank symbol 1. Indeed, $1 \oplus 1 \oplus 1 = 1$. If $C$ starts from the configuration $\ldots 11011 \ldots$, where the dots substitute 1s, then the next configuration is $\ldots 1101011 \ldots$.*

It is well-known that Turing machines can be simulated by CA with $r = 1$ and any CA can be simulated by a 2CA (with a neighborhood of a larger size).

We will need some specific requirements on these simulations; see Section 5 for more details.

**3. Sketch of the Proof**

The proof of Theorem 1 consists of the following steps.

1. Choose a suitable **EXP**-complete language $L \in \mathbf{DTIME}(n^2 2^n) \setminus \mathbf{DTIME}(2^{n/2})$ and fix a Turing machine $M$ recognizing it.
2. Construct another machine $U$ that simulates an operation of $M$ *on all inputs in parallel* (see Section 6 for more details).

3. Machine $U$ is simulated by a CA $C_U$. The cellular automaton $C_U$ is simulated in its turn by a 2CA $C_U^{(2)}$ (see Section 5 for more details) and $C_U^{(2)}$ is simulated by a $d$-dimensional FDG $\mathcal{D}_U$ (see Section 4), where $d$ depends on $C_U^{(2)}$.

4. The whole chain of simulations guarantees that the result of operating $M$ on an input $w$ (which is binary) coincides with the value of a specific position of $\mathcal{D}_U$. This position can be computed in polynomial time. Thus, we obtain a polynomial reduction of $L$ to $\mathcal{P}(D_U)$.

5. Now Theorem 1 follows from the assumption of hardness of language $L$.

## 4. From Cellular Automata to Subtraction Games

In this section, we follow Larsson and Wästlund's [19] construction with minor modifications.

### 4.1. First Step: Simulation of a 2CA by a 2-dimensional Modular Game

We are going to relate an operation of a 2CA $C = (\{0,1\}, \delta)$ starting at the configuration $c_0 = (\ldots 11011 \ldots)$ with a 2-dimensional $2N$-modular FDG $\mathcal{D}'_C$. Recall that the symbol 1 is assumed to be blank: $\delta(1, \ldots, 1) = 1$. The value of $N$ depends on $C$ and we will choose it greater than the size of a neighborhood $r$ of $C$. Our goal is to determine symbols $c_t(u)$ in the cells of $C$ by the values $p(x_1, x_2)$ of appropriate positions of $\mathcal{D}'_C$.

Basically, the coordinates $(t, u)$ and $(x_1, x_2)$ are related by a linear transform. We assume that the time arrow (the coordinate $t$) is a direction $(1, 1)$ in the coordinates $(x_1, x_2)$, while the coordinate along the automaton tape (the coordinate $u$) is in the direction $(1, -1)$.

More specifically, the configuration $c_t$ of $C$ at moment $t$ corresponds to the positions on a line $x_1 + x_2 = 2Nt$. The cell coordinate is $u = (x_1 - x_2)/2$, as shown in Figure 1 ($N = 1$). For the configuration $(\ldots 11011 \ldots)$ we assume that 0 has coordinate 0 on the automaton tape.

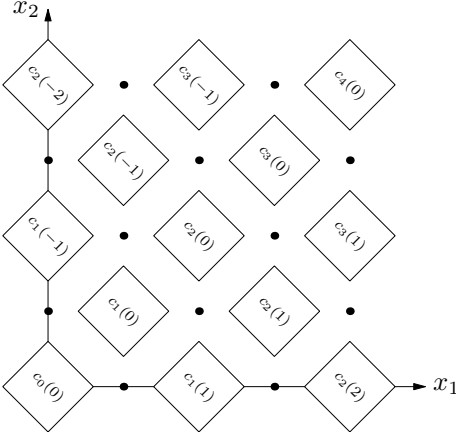

**Figure 1.** Encoding configurations of 2CA by positions of a modular FDG.

The required relation between the content of an automaton tape and the values of positions of game $\mathcal{D}'_C$ is given by the following equation:

$$c_t(u) = p(Nt + u, Nt - u) \quad \text{for } |u| \leq Nt. \tag{4}$$

The only cell 0 carries the non-blank symbol in the initial configuration. Therefore, if $|u| > Nt > rt$ then $c_t(u) = 1$. There are no positions with negative coordinates in a game. To extend the relation (4) to all values of the coordinates, we extend the value function $p(x_1, x_2)$ by the rule: $p(x_1, x_2) = 1$ if either $x_1 < 0$ or $x_2 < 0$. This rule can be explained in game terms as follows: we add dummy positions with negative values of coordinates; we regard them as terminals of value 1. It is correct, since for the game evaluation functions $[\ldots]$ the equality $[p_1, \ldots, p_k, 1, \ldots, 1] = [p_1, \ldots, p_k]$ holds. It implies that extra arguments

with the value 1 do not change the value of the function. Therefore, the dummy positions do not change the values of real positions of a game.

Under this convention, the initial configuration $c_0 = (\ldots 11011 \ldots)$ satisfies the relation (4) for any game, since the position $(0,0)$ is a P-position. To keep the relation (4) for $t > 0$, we need to specify appropriate modulus and difference sets. For this purpose, we compute the transition function of the automaton by a circuit in the basis of the Boolean functions $[p_1, \ldots, p_n]$ defined by Equation (2).

The functions $[p_1, \ldots, p_n]$ form a complete basis: any Boolean function can be computed by a circuit with gates $[\ldots]$. To prove this claim, it is sufficient to compute the functions from the standard complete basis by circuits in the basis $[\ldots]$:

$$\neg x = [x], \quad x \vee y = [[x], [y]], \quad x \wedge y = [[x, y]].$$

In the construction of difference sets we will use a circuit computing the transition function of a 2CA $C$. The circuit is a sequence of assignments $s_1, \ldots, s_N$ of the form

$$s_j := [\text{list of arguments}],$$

where arguments of the $j$th assignment may be either the input variables or the values of previous assignments $s_i$, $i < j$. The value of the last assignment $s_N$ is equal to the value of the transition function $\delta(u_{-r}, \ldots, u_{-1}, u_0, u_1, \ldots, u_r)$.

For the correctness of the construction below, we require that the arguments of the last assignment $s_N$ do not contain the input variables $u_i$. To satisfy the requirement, we assume that the circuit starts with assignments in the form $s_{i+r+1} = [u_i]$; $s_{i+3r+2} = [s_{i+r+1}]$, where $-r \leq i \leq r$. Since $u_i = s_{i+3r+2}$, the latter can be used instead of the former in the last assignment. The size $N$ of the modified circuit is obviously greater than $r$. We will construct a modular game with the modulus $2N$.

The computation by the circuit is reflected in intermediate positions of the game. More exactly, we generalize (4) and require the following relation:

$$\begin{aligned} p(Nt + i, Nt - i) &= c_t(i), \\ p(Nt + i + j, Nt - i + j) &= s_j, \quad 1 \leq j < N, \end{aligned} \tag{5}$$

where $s_j$ is the value of $j$th assignment of the circuit for values $c_t(i - r), \ldots, c_t(i), \ldots, c_t(i + r)$ of the input variables.

**Proposition 2.** *For any 2CA $C$, there exist sets $\mathcal{D}_j$ such that relation (5) holds for the values of the modular game $\mathcal{D}'_C$ with the difference sets $\mathcal{D}_j$.*

**Proof.** The choice of $\mathcal{D}_j$ depends on the arguments of assignments $s_j$.

Sets $\mathcal{D}_{2j+1}$ are irrelevant and may be arbitrary.

If an input variable $u_k$ is an argument of $s_j$ then we include in $\mathcal{D}_{2j}$ vector $(j - k, j + k)$. Since

$$(Nt + i + j, Nt - i + j) = (Nt + i + k, Nt - i - k) + (j - k, j + k),$$

it guarantees that there exists a legal move from position $(Nt + i + j, Nt - i + j)$ to position $(Nt + i + k, Nt - i - k)$.

If the value of an intermediate assignment $s_k$ is an argument of $s_j$ then we include in $\mathcal{D}_{2j}$ vector $(j - k, j - k)$. It guarantees that there exists a move from position $(Nt + i + j, Nt - i + j)$ to position $(Nt + i + k, Nt - i + k)$.

In this way we have defined the set $\mathcal{D}_{2N}$. Since $0 \equiv 2N \pmod{2N}$, we set $\mathcal{D}_0 = \mathcal{D}_{2N}$.

The rest of the proof is by induction on the parameter $A = 2Nt + 2i$, where $t \geq 0$, $0 \leq i < N$. For $A = 0$ we have $t = 0$ and $i = 0$. So the relation (5) holds as it explained above. Now suppose that the relation holds for all lines $x_1 + x_2 = A'$, $A' < 2Nt + 2j$. To

complete the proof, we should verify the relation on the line $x_1 + x_2 = 2Nt + 2j$. From the construction of the sets $\mathcal{D}_{2j}$ and the induction hypothesis we conclude that

$$p(Nt + i + j, Nt - i + j) = [\text{arguments of the assignment } s_j].$$

Here arguments of the assignment $s_j$ are the values of the input variables and the values of previous assignments in the circuit computing the transition function $\delta(c_t(u - r), \ldots, c_t(u), \ldots, c_t(u + r))$.

The last touch is to note that the value of the $N$th assignment is just the value

$$c_{t+1}(u) = \delta(c_t(u - r), \ldots, c_t(u), \ldots, c_t(u + r)). \quad \square$$

Please note that game $\mathcal{D}'_C$ has the following property: if there is a legal move from $(x_1, x_2)$ to $(y_1, y_2)$ then either $x_1 + x_2 \equiv 0 \pmod{2N}$ or the residue of $y_1 + y_2$ modulo $2N$ is less than the residue of $x_1 + x_2$. (Standardly, we assume that the residues take values $0, 1, \ldots, 2N - 1$). Note also that $x_1 + x_2 \not\equiv y_1 + y_2 \pmod{2N}$, since the input variables are not arguments of the final assignment.

**Example 5.** *Let us construct the difference sets for the 2CA from Example 4.*
*Take a circuit computing $x_{-1} \oplus x_1 \oplus 1$:*

$$s_1 := [x_{-1}, x_1] = \neg x_{-1} \vee \neg x_1,$$
$$s_2 := [x_{-1}] = \neg x_{-1},$$
$$s_3 := [x_1] = \neg x_1,$$
$$s_4 := [s_2, s_3] = x_{-1} \vee x_1,$$
$$s_5 := [s_1, s_4] = x_{-1} \wedge x_1 \vee \neg x_{-1} \wedge \neg x_1 = x_{-1} \oplus x_1 \oplus 1.$$

*So $N = 5$. Applying the construction from the proof of the proposition we get*

$$\mathcal{D}_0 = \mathcal{D}_{10} = \{(5 - 1, 5 - 1), (5 - 4, 5 - 4)\} = \{(4, 4), (1, 1)\},$$
$$\mathcal{D}_2 = \{(1 - (-1), 1 + (-1)), (1 - 1, 1 + 1)\} = \{(2, 0), (0, 2)\},$$
$$\mathcal{D}_4 = \{(2 - (-1), 2 + (-1))\} = \{(3, 1)\},$$
$$\mathcal{D}_6 = \{(3 - 1, 3 + 1)\} = \{(2, 4)\},$$
$$\mathcal{D}_8 = \{(4 - 2, 4 - 2), (4 - 3, 4 - 3)\} = \{(2, 2), (1, 1)\}.$$

*4.2. Second Step: Simulation of a 2CA by a $(2N + 2)$-Dimensional Subtraction Game*

To exclude modular conditions, we use the trick suggested in [19].

For a 2CA $C$ we have defined the 2-dimensional $2N$-modular game $\mathcal{D}'_C$ in the previous subsection. To join all difference sets together we add $2N$ more coordinates and realize a counter modulo $2N$ on the unit coordinate vectors in the additional coordinates. Specifically, we construct a $(2N + 2)$-dimensional FDG $\mathcal{D}_C$ with the difference set

$$\mathcal{D} = \left\{ (a_1, a_2, 0^{2N}) + e^{(j)} - e^{(k)} : (a_1, a_2) \in D_j, \ k = j - a_1 - a_2 \pmod{2N} \right\}.$$

Here $e^{(i)}$ is the $(i + 2)$th coordinate vector: $e^{(i)}_{i+2} = 1$, $e^{(i)}_s = 0$ for $s \neq i + 2$.

**Proposition 3.** *The value of a position $(x_1, x_2, 0^{2N}) + e^{(2r)}$ of the game $\mathcal{D}_C$ is equal to the value of a position $(x_1, x_2)$ of the modular game $\mathcal{D}'_C$ if $2r \equiv x_1 + x_2 \pmod{2N}$.*

**Proof.** We prove the proposition by induction on $t = x_1 + x_2$. The base case $t = 0$ holds, by the convention on the values of dummy positions (with negative coordinates).

The induction step. A legal move at a position $(Nt + i + j, Nt - i + j, 0^{2N}) + e^{(2j)}$ is to a position $(Nt + i + j, Nt - i + j, 0^{2N}) - (a_1, a_2, 0^{2N}) + e^{(2s)}$, where $2s \equiv 2j - a_1 - a_2$

(mod $2N$) and $(a_1, a_2) \in \mathcal{D}_{2j}$. It corresponds to a move from $(Nt + i + j, Nt - i + j)$ to $(Nt + i + j - a_1, Nt - i + j - a_2)$ in the modular game. □

From Propositions 2 and 3 we derive:

**Corollary 1.** *For any 2CA C there exist an integer N and a $(2 + 2N)$-dimensional FDG $\mathcal{D}_C$ such that the relation*

$$c_t(u) = p(Nt + u, Nt - u, 0, 0, \ldots, 0, 1) \quad \text{holds for } |u| \leq Nt.$$

## 5. From Turing Machines to Cellular Automata

In this section, we describe how to simulate a Turing machine by a binary cellular automaton. It is standard, except for some specific requirements.

Let $M = (Q, \{0, 1\}, \Gamma, \delta_M, 1, 2, 3)$ be a Turing machine, where the input alphabet is binary, $Q = \{1, \ldots, q\}$, $q \geq 3$ is the set of states, $\Gamma = \{0, 1, \ldots, \ell\}$ is the tape alphabet, $\ell > 1$ is the blank symbol, $\delta_M \colon Q \times \Gamma \to Q \times \Gamma \times \{+1, -1\}$ is the transition function, and $1, 2, 3$ are the start, accept, and reject states, respectively. The tape of the machine assumed to be infinite in both directions.

A configuration of $M$ specifies the content of the tape, the position of the head and the state of the machine. We encode the configuration by a doubly infinite string $c \colon \mathbb{Z} \to A$, where the set $A = \{0, \ldots, q\} \times \{0, \ldots, \ell\}$ used to combine the content of a cell and the information about the head. Please note that $0 \notin Q$. So, we assume that the head position is indicated by a pair $(q, a)$, $q > 0$, $a \in \Gamma$; the content of any other cell is encoded as $(0, a)$, $a \in \Gamma$.

An operation of $M$ starting from the configuration $c_0$ generates a sequence of encoded configurations $c_0, \ldots, c_t, \ldots$. It is easy to see that $c_{t+1}(u)$ is determined by $c_t(u - 1)$, $c_t(u)$, $c_t(u + 1)$. In other words, there is a function $\delta_C \colon A^3 \to A$ such that $c_{t+1}(u) = \delta(c_t(u - 1), c_t(u), c_t(u + 1))$ for all $u$ and $t \geq 0$. Thus, the CA $C_M = (A, \delta_C)$ over the alphabet $A$ simulates the operation of $M$ in encoded configurations. The symbol $\Lambda_C = (0, \ell)$ is the blank symbol: $\delta_C(\Lambda_C, \Lambda_C, \Lambda_C) = \Lambda_C$ (indeed, $\ell$ is the blank symbol of $M$, and $0$ in the first coordinate indicates that the head is not over the cell).

The next step is to simulate $C_M$ by a 2CA $C_M^{(2)}$. The idea of simulation is simple: just a binary encoding of symbols from $A$. But in the chain of reductions outlined in Section 3 we rely on specific features of the simulation. Therefore, we provide a detailed formal definition.

Let $C_M' = (A', \delta_C')$ be an automaton isomorphic to $C_M$, where $A' = \{0, \ldots, L - 1\}$ and $L = |A| = (|Q| + 1) \cdot |\Gamma|$. The transition function $\delta_C'$ of the automaton is defined as follows

$$\delta_C'(i, j, k) = \pi(\delta_C(\pi^{-1}(i), \pi^{-1}(j), \pi^{-1}(k))),$$

where $\pi \colon A \to A'$ is a bijection. To keep the relation between the start configurations, we require that $\pi(\Lambda_C) = 0$, $\pi((1, \ell)) = 1$. Recall that $1$ is the start state of $M$ and $\ell$ is the blank symbol of $M$.

The symbols of $A'$ are encoded by binary words as follows

$$\varphi(a) = 1^{1+L-a} 0^a 1.$$

Thus, each symbol is encoded by a word of length $L + 2$. In particular, $\varphi(0) = \varphi(\pi(\Lambda_C)) = 1^{L+2}$ and $\varphi(1) = \varphi(\pi(1, \ell)) = 1^L 01$. The encoding $\varphi$ is naturally extended to words in the alphabet $A'$ (finite or infinite). The extended encoding is also denoted by $\varphi$.

Thus, the start configuration of $M$ with the empty tape corresponds to the configuration $\ldots 1110111 \ldots$ of $C_M^{(2)}$. Recall that $1$ is the blank symbol of $C_M^{(2)}$.

To construct the transition function of $C_M^{(2)}$, we use the following alignment of configurations: if $i = q(L + 2) + k$, $0 \leq k < L + 2$ then $\varphi(c)(i)$ is the $(k + 1)$th bit of $\varphi(c(q))$.

The size of a neighborhood of $C_M^{(2)}$ is $r = 2(L + 2)$. To define the transition function $\delta_M^{(2)}$, we use a local inversion property of the encoding $\varphi$: looking at the $r$-neighborhood of

an $i$th bit of $\varphi(c)$, where $i = q(L + 2) + k$, $0 \le k < L + 2$, one can restore symbols $c(q - 1)$, $c(q)$, $c(q + 1)$, and position $k$ of the bit provided the neighborhood contains zeroes (0 is the non-blank symbol of $C_M^{(2)}$).

**Example 6.** *Let $L = 4$, $r = 12$. Assume that the neighborhood of the i-th bit, for some unknown i, is*

$$11111001110\underline{0}11100011111.$$

*The series of three zeroes containing the i-th bit (underlined) is surrounded by ones. Therefore, it is a part of the symbol code $\varphi(3) = 110001$. It gives us a decomposition of the neighborhood into symbol codes:*

$$11\ 111001\ 110\underline{0}01\ 110001\ 1111.$$

*Thus, $c(q - 1) = 2$, $c(q) = 3$, $c(q + 1) = 3$, $k = 3$ (the fourth bit of the symbol code). Of course, we cannot restore the value q and the absolute position i of the bit.*

Please note that if the $r$-neighborhood of a bit does not contain zeroes then the bit is a part of encoding of the blank symbol 0 of $C_M'$ and, moreover, $c(q - 1) = c(q) = c(q + 1) = 0$.

We generalize the arguments from the above example to prove the following fact.

**Lemma 1.** *There exists a function $\delta_C^{(2)}: \{0, 1\}^{2r+1} \to \{0, 1\}$ such that a 2CA $C_M^{(2)} = (\{0, 1\}, \delta_C^{(2)})$ simulates $C_M'$: starting from $b_0 = \ldots 1110111 \ldots$, it produces the sequence of configurations $b_0, b_1, \ldots$ such that $b_t = \varphi(c_t)$ for any t, where $(c_t)$ is the sequence of configurations produced by $C_M'$ starting from the configuration $c_0 = \ldots 0001000 \ldots$*

**Proof.** The function $\delta_C^{(2)}$ should satisfy the following property. If $b = \varphi(c)$ then

$$\delta_C^{(2)}\big((b(i - r), \ldots, b(i), \ldots, b(i + r)\big) = \varphi\big(\delta_C'(c(q - 1), c(q), c(q + 1))\big)(k) \tag{6}$$

for all integers $i = q(L + 2) + k$, $0 \le k < L + 2$. This property means that applying the function $\delta_C^{(2)}$ to $b$ produces the configuration $b_1 = \varphi(c_1)$, where $c_1$ is the configuration produced by the transition function $\delta_C'$ from the configuration $c$. Therefore, the sequence of configurations produced by $C_M^{(2)}$ starting at $\varphi(c_0)$ is the sequence of the encodings of configurations $c_t$ produced by $C_M'$ starting at $c_0$.

Please note that $\varphi(c(q - 1))$, $\varphi(c(q))$ and $\varphi(c(q + 1))$ are in the $r$-neighborhood of a bit $i$.

Thus, from the condition on blank symbols in the alphabets $A'$ and $\{0, 1\}$, we conclude that the required property holds if the $r$-neighborhood of a bit $i$ does not contain zeroes (non-blank symbols of $C_M^{(2)}$). In this case, the $i$th bit of $b$ is a part of the encoding of the blank symbol 0 in the alphabet $A'$ and, moreover, $c(q - 1) = c(q) = c(q + 1) = 0$.

Now suppose that the $r$-neighborhood of the $i$th bit contains zeroes. Take the nearest zero to this bit (either from the left or from the right) and the maximal series $0^a$ containing it. The series is a part of the encoding of a symbol in $c$. Therefore, there are at least $1 + L - a$ ones to the left of it. They all are contained in the $r$-neighborhood of the bit. Thus, we locate an encoding of a symbol $c(q + q')$, $q' \in \{-1, 0, +1\}$, and we are able to determine $q'$. It depends on relative position of the $i$th bit with respect to the first bit $j$ of the symbol located. If $i - j \ge L + 2$ then $q' = -1$; if $L + 2 > j - i \ge 0$ then $q' = 0$; otherwise, $i - j < 0$ and $q' = 1$. Therefore, the symbols $c(q - 1)$, $c(q)$, $c(q + 1)$ can be restored from the $r$-neighborhood of the $i$th bit. Moreover, a relative position $k$ of the $i$th bit in $\varphi(c(q))$ can also be restored.

Because the symbols $c(q - 1)$, $c(q)$, $c(q + 1)$ and the position $k$ are the functions of the $r$-neighborhood of the bit $i$, it is correct to define the function $\delta_C^{(2)}$ as

$$\delta_C^{(2)}(u_{-r}, \ldots, u_0, \ldots, u_r) = \varphi\big(\delta_M'(c(q - 1), c(q), c(q + 1))\big)(k)$$

if the restore process is successful on $(u_{-r}, \ldots, u_0, \ldots, u_r)$; for other arguments, the function can be defined arbitrarily. It is clear that this function satisfies the property (6). □

## 6. Parallel Execution of a Turing Machine

In our plan of the proof (see Section 3), we simulate an operation of a specific Turing machine $M$ on a specific input $w$ by an operation of a 2CA on the fixed starting configuration (as it described in the previous section). Thus we need yet another construction: a Turing machine $U$ simulating an operation of a Turing machine *M on all inputs*. The idea of simulation is well-known, but again, we need to specify some details of the construction.

We use notation from the previous section and assume that on each input of length $n$ the machine $M$ makes at most $T(n)$ steps, where $T(n) > n$. It is convenient to include in the alphabet of $U$ the set $A = \{0, \ldots, q\} \times \{0, \ldots, \ell\}$. Note in advance that there are some additional symbols in the alphabet.

An operation of $U$ is divided into *stages* while its tape is divided into *zones*. Each zone is used to simulate an operation of $M$ on a specific input $w$. All the zones are placed between delimiters, say, $\triangleleft$ and $\triangleright$. In calculations below we assume that $\triangleleft$ is placed in the cell 0. Also, the zones are separated by a delimiter, say, $\diamond$.

At the start of a stage $k > 1$ there are $k - 1$ zones corresponding to the inputs $w_1, w_2, \ldots, w_{k-1}$ of $M$. We order inputs (i.e., binary words) by their lengths and words of equal length are ordered lexicographically.

Each zone consists of three blocks. The blocks in a zone are separated by a delimiter, say #. They are pictured in Figure 2.

| 1 | | $n$ | | $n + 2T(n)$ |
|---|---|---|---|---|
| result | # | input | # | work place |

**Figure 2.** A zone on the tape of $U$.

Machine $U$ maintains the following content in the blocks at the start of a stage $k > 1$. The content of the second block of a zone $i$ specifies the input $w_i$ of the machine $M$. The size of the first block is 1. The cell in this block carries the symbol yes iff $M$ accepts the input $w_i$. Otherwise, it carries the symbol no. In this way we do not distinguish an unfinished computation and the negative result. However, it does not affect the correctness of reductions built on this construction. The last block of the zone contains a configuration of $M$ after running $k - 1 - i$ steps on input $w_i$. In particular, the last zone contains the initial configuration on input $w_{k-1}$. The configurations are represented by words over the alphabet $A$, as described in Section 5.

During the first stage, machine $U$ writes on the tape the first zone containing the initial configuration on the input $w_1$. The zone is surrounded by $\triangleleft$ and $\triangleright$.

During the stage $k > 1$, machine $U$ traverses the tape from the left delimiter $\triangleleft$ to the right delimiter $\triangleright$. Entering a zone, it simulates the next step of operation of $M$ on the corresponding input. At the end of the stage machine $U$ creates a new zone corresponding to the input $w_k$ and the initial configuration of $M$ on this input. The initial configuration is extended in both directions by white space of size $T(n)$, see Figure 3.

| 1 | | $n$ | | $n + 2T(n)$ |
|---|---|---|---|---|
| no | # | $w_k$ | # | $(0, \ell) \ldots (0, \ell)(1, w_{k,1})(0, w_{k,2}) \ldots (0, w_{k,n})(0, \ell) \ldots (0, \ell)$ |

**Figure 3.** A fresh zone on the stage $k$.

If the simulation procedure detects that the operation of $M$ on an input $w_i$ is finished, machine $U$ updates the resulting block if necessary and it does not change the zone on the subsequent stages.

In the arguments below we need $U$ to satisfy specific properties.

**Proposition 4.** *If $T(n) = C \cdot (n+1)^2 \cdot 2^n$ for some integer constant $C \geq 1$ then there exists $U$ operating as is described above such that*

1.  *For all sufficiently large $n$ machine $U$ produces the result of operation of $M$ on the input $w$ of length $n$ in time $< 2^{4n}$.*
2.  *The head of $U$ visits the first blocks of zones only on steps $t$ that are divisible by 3.*

**Proof.** At first, we show how to construct a machine $U'$ satisfying the property 1. More exactly, we explain how to construct a machine satisfying the following claims.

*Claim 1.* Updating a configuration of the simulated machine $M$ into a zonetakes a time $O(S)$, where $S$ is the size of the zone.

A straightforward way to implement the update is the following. The head of $U'$ scans the zone until it detects a symbol $(q, a)$ with $q > 0$. It means that the head of the simulated machine $M$ is over the current cell. Then $U'$ updates the neighborhood of the cell detected with respect to the transition function of $M$. If $q$ is a final state then no changes are made. After that $U'$ continues a motion until it detects the next zone.

If machine $M$ finishes its operation on a configuration written in the current zone and the final state is accept then additional actions should be done. Machine $U'$ should update the resulting block. For this purpose it returns to the left end of the zone, updates the result block and continues a motion to the right until it detects the next zone.

In this way, each cell in the zone is scanned $O(1)$ times. The total time for update is $O(S)$.

*Claim 2.* A fresh zone on the stage $k$ is created in time $O(nT(n))$, where $n = |w_k|$.

Creation of the resulting block takes a time $O(1)$.

To compute the next input word machine $U'$ copies the previous input into the second block of the fresh zone. The distance between positions of the second blocks is $O(1) + |w_{k-1}| + 2T(|w_{k-1}|) = O(T(n))$. Here the term $O(1)$ counts delimiters between the blocks. Machine $U'$ should copy at most $n$ symbols. Therefore, the copying takes a time $O(nT(n))$.

After that, machine $U'$ computes the next word in the lexicographical order. It can be done by adding 1 modulo $2^{|w_{k-1}|}$ to $\mathrm{bin}(w_{k-1})$, where $\mathrm{bin}(w)$ is the integer represented in binary by $w$ (the empty word represents 0). It requires a time $O(n)$. If an overflow occurs, then the machine should write an additional zero. It also requires a time $O(n)$.

To mark the third block in the fresh zone machine $U'$ computes a binary representation of $T(n)$ by a polynomial time algorithm using the second block as an input to the algorithm (thus, $n$ is given in unary). Then it makes $T(n)$ steps to the right using the computed value as a counter and decreasing the counter each step. The counter should be moved along a tape to save time. The length of binary representation of $T(n)$ is $O(n)$. Therefore, each step requires $O(n)$ time and totally marking of $T(n)$ free space requires $\mathrm{poly}(n) + O(nT(n)) = O(nT(n))$ time.

Then $U'$ copies the input word $w_k$ to the right of marked free space. It requires $O(nT(n))$ time. The first cell of the copied word should be modified to indicate the initial state of the simulated machine $M$. In addition, finally, it repeat the marking procedure to the right of the input.

The overall time is $O(nT(n))$.

Let us prove that the property 1 is satisfied by the machine $U'$. Counting time in stages, the zone corresponding to an input word $w$ of length $n$ appears after $\leq 2^{n+1}$ stages. After that, the result of operation of $M$ appears after $\leq T(n)$ stages.

Let $s = |w_k|$. At stage $k$ there are at most $2^{s+1}$ zones. Updating the existing zones requires time $O(2^s(s + T(s)))$ due to Claim 1. Creation of a fresh zone requires time $O(sT(s))$ due to Claim 2. Thus, the overall time for a stage is

$$O\big(2^{s+1}(s + T(s)) + sT(s)\big) = O(2^s T(s)).$$

Therefore, the result of operation of $M$ appears in time

$$O\big((2^{n+1} + C \cdot (n+1)^2 \cdot 2^n) \cdot 2^n \cdot C \cdot (n+1)^2 \cdot 2^n\big) = O(n^4 2^{3n}) < 2^{4n} \tag{7}$$

for sufficiently large $n$.

Now we explain how to modify the machine $U'$ to satisfy the property 2. Please note that the resulting block of a zone is surrounded by delimiters: # to the right of it and either ◁ or ◇ to the left.

We enlarge the state set of $U'$ adding a counter modulo 3. It is increased by $+1$ at each step of operation. If the head of the modified machine $U$ is over the ◁ or ◇ and $U'$ should go to the right, then machine $U$ makes dummy moves in the opposite direction and back to ensure that it visits the cell to the right on a step $t$ divisible by 3. In a similar way machine $U$ simulates the move to the left from the cell carrying the delimiter #.

The running time after this modification is multiplied by a factor $O(1)$. Thus, after the modification the last inequality in Equation (7) holds for sufficiently large $n$. □

## 7. Proof of the Main Theorem

As a hard language, we use the language of the bounded halting problem

$$L = \{(\langle M \rangle, x, 0^k) : M \text{ accepts } x \text{ after at most } 2^k \text{ steps}\},$$

which is obviously **EXP**-complete. The following proposition is an easy corollary of the time hierarchy theorem [25].

**Proposition 5.** *$L \notin$ **DTIME**$(2^{n/2})$, where n is the input size.*

**Proof.** Take an arbitrary language $L' \in$ **DTIME**$(2^{2n})$, where $n$ is the input size. Let $M'$ be a Turing machine recognizing $L'$ in time $2^{2n+c}$, where $c$ is a suitable constant. The function

$$x \mapsto (\langle M' \rangle, x, 0^{2|x|+c})$$

provides a polynomial reduction of $L'$ to $L$.

Let us suppose for the sake of contradiction that $L$ is recognized by a Turing machine $M$ in time $C \cdot 2^{n/2}$, where $n$ is the input size. Consider an algorithm to decide $x \in L'$:

1.  Compute $(\langle M' \rangle, x, 0^{2|x|+c})$ (requires poly$(|x|)$ time).
2.  Run $M$ on the input $(\langle M' \rangle, x, 0^{2|x|+c})$ (requires $O(2^{3|x|/2})$ time).

The total running time of this algorithm is $O(2^{1.5n})$.

Thus, **DTIME**$(2^{2n}) \subseteq$ **DTIME**$(2^{1.5n})$. It contradicts the time hierarchy theorem. □

Using universal Turing machine and counters it is easy to get an upper bound of the algorithmic complexity of $L$.

**Proposition 6.** *$L \in$ **DTIME**$(n^2 \cdot 2^n)$, where n is the input size.*

**Proof.** To verify $(\langle M \rangle, x, 0^k) \in L$ one needs to implement a universal Turing machine and a counter of size $k$. To speed up computation, the description $\langle M \rangle$ and the counter are shifting along the tape. Shifting by 1 requires time $O(|\langle M \rangle| + k))$ as well as updating the counter. To apply the transition function of $M$, it suffices to align the corresponding entry of the transition function with the current position of the head.

Thus, to simulate a step of computation made by $M$ on the input $x$, one needs $O((|\langle M \rangle| + k)^2) = O(n^2)$ time, where $n$ is the input size.

The total number of simulation steps is at most $2^k \leq 2^n$. □

Based on Proposition 6, we fix a Turing machine $M$ recognizing $L$ such that for some constant $C$ the machine $M$ makes at most $T(n) = C \cdot (n+1)^2 \cdot 2^n$ steps on inputs of size $n$.

For machine $M$ apply the construction from Section 6 and Proposition 4 to construct the machine $U$. Then convert $U$ into 2CA $C_U^{(2)}$ as described in Section 5. We put an additional requirement on bijection $\pi$, namely $\pi(0, \text{yes})) = L - 1$. It locates the result of computation of $M$ in the third bit of the encoding of the result block. More specifically, this bit is 0 iff $M$ accepts the corresponding input. Finally, construct $O(1)$-dimensional FDG $\mathcal{D}_C$ as it described in Section 4. The dimension $2N + 2$ of the game is determined by machine $M$. By Corollary 1, the symbol $c_t(u)$ on the tape of $C_U^{(2)}$ equals the value of position $(Nt + u, Nt - u, 0, 0, \ldots, 0, 1)$ of the game.

Define the function $\rho\colon w \mapsto (Nt + u, Nt - u, 0, 0, \ldots, 0, 1)$ as follows. Set $t = 2^{4n}$, where $n$ is the size of $w$. Set $u$ be the position of the bit carrying the result of computation of $M$ on input $w$ in the image of the resulting block of the zone $k$ corresponding to the input $w$.

**Proposition 7.** (i) There exists a polynomial reduction of the language $L$ to the language $\mathcal{P}(\mathcal{D})$ based on the function $\rho$; (ii) $u = O(2^{3n})$.

**Proof.** The reduction is defined by the function

$$\rho\colon w \mapsto (Nt + u, Nt - u, 0, 0, \ldots, 0, 1)$$

for sufficiently large inputs (taking into account Proposition 4). For the rest of inputs (there are $O(1)$ of them), the reduction is defined in an arbitrary way preserving correctness.

Correctness of this reduction is ensured by constructions from Sections 4–6 and by Proposition 4.

To compute the function $\rho\colon w \mapsto (Nt + u, Nt - u, 0, 0, \ldots, 0, 1)$ one needs to compute two values.

1. The number $k$ of the zone corresponding to the input $w$ of length $n$.
2. The position $u$ of the bit carrying the result of computation of $M$ on input $w$ in the image of the resulting block of the zone $k$.

We are going to prove that computations on step 1 and 2 require time polynomial in $n$ (part (i) of the proposition) and $u = O(2^{3n})$ (part (ii)).

For the first claim, note that $k = 2^n + \text{bin}(w)$ (recall that $\text{bin}(w)$ is the integer represented in binary by $w$). Indeed, there are $2^n - 1$ shorter words, all of them precede $w$ in the ordering of binary words that we use. Also, there are exactly $\text{bin}(w)$ words of length $n$ preceding the word $w$. The formula for $k$ follows from these observations (note that we count words starting from 1).

It is quite obvious now that $k$ is computed in polynomial time.

For the second claim, we should count the sizes of zones preceding the zone for $w$ and add a constant to take into account delimiters. Let count the size of a zone including the delimiter to the left of it. Then the size of a zone for an input word of length $\ell$ is

$$1 + 1 + 1 + \ell + 1 + \ell + 2T(\ell) = 4 + 2\ell + 2T(\ell).$$

There are $2^\ell$ words of length $\ell$. Thus, the total size of the zones preceding the zone of $w$ is

$$S = \sum_{\ell=0}^{n-1} 2^\ell (4 + 2\ell + 2T(\ell)) + \text{bin}(w)(4 + 2n + 2T(n)) + 2.$$

For $T(n) = C \cdot (n+1)^2 \cdot 2^n$ this expression can be computed in polynomial time in $n$ by a straightforward procedure (the expression above has $\text{poly}(n)$ arithmetic operations and the results of these operations are integers represented in binary by $\text{poly}(n)$ bits).

Thus, the resulting block of the zone of $w$ is $S + 1$ (the delimiter to the left of the zone adds 1).

To compute $u$ we should multiply $S + 1$ by $L + 2 = O(1)$ (the size of encoding) and add 3 (because the third bit indicates the result of computation of the simulated machine $M$).

All these calculations can be done in polynomial time.

Now the required upper bound on $u$ can be easily verified:

$$u \leq (L+2) \cdot \left( n2^n(4 + 2n + 2C(n+1)^2 2^n) + 2^n(4 + 2n + 2C(n+1)^2 2^n) + 3 \right) + 3 = O(2^{3n}).$$

$\square$

Now the first claim of Theorem 1 follows from part (i) of Proposition 7. For the second claim, we use the part (ii) of Proposition 7. It implies that the reduction maps a word of size $n$ to a vector of size at most $8n + O(1)$. So, if an algorithm $\mathcal{A}$ solves the game $\mathcal{D}_C$ in time $T_{\mathcal{A}}(m)$, then the composition of the reduction and $\mathcal{A}$ recognizes $L$ in time $\mathrm{poly}(n) + T_{\mathcal{A}}(8n + O(1))$. By Proposition 5 this value is $\Omega(2^{n/2})$. We conclude that $T_{\mathcal{A}}(m) = \Omega(2^{m/16})$.

## 8. Discussion

Our hardness result drastically contrasts with the results on slow nim in [13,14]. For these games, a rather simple formula was obtained for the SG function. So, these games are solved efficiently.

Slow versions of nim can be considered to be FDG, see Example 1. The difference sets for these games consist of $(0, 1)$-vectors.

A more general subclass of FDG is formed by games whose difference vectors are nonnegative. In other words, players are not allowed to add pebbles to heaps.

In our proof we use the construction by Larsson and Wästlund. The negative coordinates of difference vectors are essential in it. We see no way to maintain a counter modulo $N$ without such vectors in the difference set.

Moreover, we believe that the existence of difference vectors with negative coordinates is essential for both hardness results: ours and the undecidability result from [19].

**Conjecture 1.** *For any FDG game such that $a_i \geq 0$ for each $(a_1, \ldots, a_d) \in \mathcal{D}$, there exists a polynomial algorithm solving the game, i.e., $\mathcal{P}(\mathcal{D}) \in \mathbf{P}$.*

To the best of our knowledge, P-positions of all efficiently solved FDG form semilinear sets. Recall that a *semilinear set* is a set of vectors from $\mathbb{N}^d$ that can be expressed in Presburger arithmetic; see, e.g., [26]. Presburger arithmetic admits quantifier elimination. Therefore, a semilinear set can be expressed as a finite union of solutions of systems of linear inequalities and equations modulo some integer (fixed for the set).

**Conjecture 2.** *For any FDG game such that $a_i \geq 0$ for each $(a_1, \ldots, a_d) \in \mathcal{D}$, the set of P-positions is semilinear.*

It is well-known that evaluating a formula in Presburger arithmetic is a very hard problem. The first doubly exponential time bound for this problem was established by Fischer and Rabin [27]. Berman proved that the problem requires doubly exponential space [28].

However, these hardness results are irrelevant for our needs. In solving FDG games we deal with sets of dimension $O(1)$ and we may hardwire the description of the set of P-positions in an algorithm solving the game. Verifying a linear inequality takes a polynomial time as well as verifying an equation modulo some integer. Thus, any semilinear set of dimension $O(1)$ can be recognized by a polynomial time algorithm. Therefore Conjecture 2 implies Conjecture 1.

Conjecture 2 also implies that the equivalence problem for FDG with nonnegative difference vectors is decidable, since Presburger arithmetic is decidable. Indeed, by Post's

theorem a set is decidable if it is enumerable and co-enumerable. The set $E = \{(\mathcal{D}_1, \mathcal{D}_2) : \mathcal{P}(\mathcal{D}_1) = \mathcal{P}(\mathcal{D}_2)\}$ is trivially co-enumerable, since it is decidable to check whether $x \in \mathcal{P}(\mathcal{D}_1) \setminus \mathcal{P}(\mathcal{D}_2)$ applying the kernel construction algorithm. On the other hand, the set $\{(\langle S \rangle, \mathcal{D}_1, \mathcal{D}_2) : S = \mathcal{P}(\mathcal{D}_1) = \mathcal{P}(\mathcal{D}_2)\}$, where $\langle S \rangle$ is a description of a semilinear set $S$, is decidable: the statement $S = \mathcal{P}(\mathcal{D})$ can be expressed in Presburger arithmetic. Therefore set $E$ is enumerable.

Why do we believe in Conjecture 2 despite results from [19] and ours? It seems that some inductive arguments can be applied in the case of nonnegative difference vectors. Please note that Conjecture 2 holds for a 1-dimensional FDG [22]. If a coordinate becomes zero it remains zero during the rest of the game (if all difference vectors are nonnegative). Thus, it might be possible to prove the implication: if "boundary conditions" are semilinear then the solution of a game is also semilinear. (After the paper was submitted, Michael Raskin pointed out that this claim is false for *arbitrary* semilinear boundary conditions. However, his arguments cannot be applied to boundary conditions produced by games). The above observations show a way to Conjecture 2.

**Author Contributions:** The authors contributed equally to this work. All authors have read and agreed to the published version of the manuscript.

**Funding:** This research was funded by RSF grant number 20–11–20203.

**Acknowledgments:** The authors are grateful to the anonymous referee of the CSR conference for several helpful remarks improving both the results and their presentation.

**Conflicts of Interest:** The authors declare no conflict of interest.

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
