# Peer review of "On Computational Hardness of Multidimensional Subtraction Games†"

_algorithms, doi:10.3390/a14030071_

Round 1

Reviewer 1 Report

This paper studies the computational complexity of a two-player game called subtraction game. A position in this game is represented by a $d$-dimensional vector with non-negative coordinates. The rules are the following: the players, alternately, have to subtract a vector, chosen from an allowable set (the difference set), from the vector representing the current position without creating negative coordinates. This action is called a move. The player who has no move left loses the game. In general, a vector in the difference set is allowed to have negative coordinates. However, to ensure the finiteness of the game, the sum of all coordinates has to be strictly positive.   Subtraction games generalize several variants of the classical game of nim and admit an efficient algorithm for the case of $d=1$. However, the complexity for higher values of $d$ are currently unknown. This paper shows that, in general, solving a subtraction game is EXP-complete.   The reduction makes use of standard arguments (ideas from [19], simulation of a Turing Machine through a Cellular Automata, Parallel simulation of a Turing Machine, Universal Turing Machines and counters) arranged in a proper way. Although not particularly innovative, the paper provides an answer to a natural question and also poses some interesting open problems that may foster further investigation.   The quality of writing is fairly good.   Minor Comments:   - in the Introduction, some additional explanations for non-expert readers should be provided. At lines 13 and 14, you mention the SG function of the disjunctive compound. As this function is widely cited in the subsequent paragraphs, an intuitive (not necessarily precise) definition would be welcome. Similarly, at line 27, P-positions are mentioned without defining what they are.   - line 71: which -> whose   - Equation (1): G(X) -> SG(X) and G(y) -> SG(y)   - Equation (2): the last ( is misplaced   - line 103: solving of -> solving   - line 113: to subtract vectors $a^{(v)}$ and $a^{(u)}$, where $(u,v)\in E$ -> to subtract any vector $a^{(u)}$ such that $(u,v)\in E$   - line 131: posiible -> possible   - line 232: what is \Lambda?   - line 258: This -> Thus   - line 412: what do you mean by bin(w)?   - line 417: a full stop is missing at the end of the previous equation   - line 475: both, -> both

Author Response

- in the Introduction, some additional explanations for non-expert readers should be provided. At lines 13 and 14, you mention the SG function of the disjunctive compound. As this function is widely cited in the subsequent paragraphs, an intuitive (not necessarily precise) definition would be welcome. Similarly, at line 27, P-positions are mentioned without defining what they are.
Response: we added brief explanations to the Introduction. The SG function and P-positions are introduced in the same place.

- line 71: which -> whose - Equation (1): G(X) -> SG(X) and G(y) -> SG(y) - Equation (2): the last
( is misplaced
- line 103: solving of -> solving
Response: fixed

- line 113: to subtract vectors $a^{(v)}$ and $a^{(u)}$, where $(u,v)\in E$ -> to subtract any vector $a^{(u)}$ such that $(u,v)\in E$
Response: edited. We leave the vector $a^{(v)} because it can not be subtracted also. It is important for isomorphism of the games.

- line 131: posiible -> possible
Response: fixed

- line 232: what is \Lambda?
Response: it was a typo. \Lambda should indicate the blank (and the correct name is $\ell$ in our notation). But Sipser does not put the blank into the tuple representing TM. So, it is just deleted. 

- line 258: This -> Thus
Response: fixed

- line 412: what do you mean by bin(w)?
Response: an integer represented by a binary word. A reminder was added.

- line 417: a full stop is missing at the end of the previous equation
- line 475: both, -> both
Response: fixed

Reviewer 2 Report

The authors describe the rather extensive literature on "subtraction games" such as Nim and variants. Some of the work that is cited is quite recent, indicating that this is an area of active research, especially concerning the question of describing the boundary between the cases where there is an algorithm to determine a winning strategy, and the cases where this question is undecidable. In this work, the authors look at the computational complexity of these problems.

They present a particular game in this class that is EXP-complete.  This matches the upper bound for games in this class, and thus the complexity bound for the class is tight.

I found only minor problems with English in the submission, and I found no technical errors.  I recommend publication, where authors will be able to incorporate the minor suggestions below with no further interaction with me.

Line 14: change "Informally, SG" to "Informally, the SG".

Line 47: change "case of fixed number" to "case of a fixed number".

Line 105: change "if x_i are binary" to "if the x_i are presented in binary notation".

Line 135: change "Then possible" to "Then the possible".

Line 140: change "for reader's" to "for the reader's".

Line 164: change "fore" to "for".

Line 171: change "Theorem follows" to "Theorem 1 follows".

Line 279: change "part of encoding" to "part of the encoding".

Line 290: change "arbitrary" to "arbitrarily".

Line 320: change "as it" to "as is".

Lines 436-437: change "games rather" to "games, a rather".

Line 445: change "that existence" to "that the existence".

Line 456: change "evaluating formula" to "evaluating a formula".

Line 460: change "and may" to "and we may".

Lilne 461: change "modulo" to "modulo some".

Line 465: change "the Post theorem" to "Post's theorem".

Author Response

We have made all changes proposed.

Line 14: change "Informally, SG" to "Informally, the SG".
Answer: "Informally, SG" is changed to "Informally, the SG".

Line 47: change "case of fixed number" to "case of a fixed number".
Answer: "case of fixed number" is changed to "case of a fixed number".

Line 105: change "if x_i are binary" to "if the x_i are presented in
binary notation".
Answer: "if x_i are binary" is changed to "if the x_i are presented in
binary notation".

Line 135: change "Then possible" to "Then the possible".
Answer: "Then possible" is changed to "Then the possible".

Line 140: change "for reader's" to "for the reader's".
Answer: "for reader's" is changed to "for the reader's".

Line 164: change "fore" to "for".
Answer: "fore" is changed to "for".

Line 171: change "Theorem follows" to "Theorem 1 follows".
Answer: "Theorem follows" is changed to "Theorem 1 follows".

Line 279: change "part of encoding" to "part of the encoding".
Answer:  "part of encoding" is changed to "part of the encoding".

Line 290: change "arbitrary" to "arbitrarily".
Answer: "arbitrary" is changed to "arbitrarily".

Line 320: change "as it" to "as is".
Answer: "as it" is changed to "as is".

Lines 436-437: change "games rather" to "games, a rather".
Answer: "games rather" to "games, a rather".

Line 445: change "that existence" to "that the existence".
Answer:  "that existence" is changed to "that the existence".

Line 456: change "evaluating formula" to "evaluating a formula".
Answer:  "evaluating formula" is changed to "evaluating a formula".

Line 460: change "and may" to "and we may".
Answer:  "and may" is changed to "and we may".

Line 461: change "modulo" to "modulo some".
Answer: "modulo" is changed to "modulo some".

Line 465: change "the Post theorem" to "Post's theorem".
Answer: "the Post theorem" is changed to "Post's theorem".

Reviewer 3 Report

The authors of this paper introduce and study a class FDG of subtraction games. An FDG game is a 2-player impartial game defined by a finite set D of vectors each of which contains d integers that sum up to a strictly positive value (the positivity condition). Each configuration of the game is described by a d-dimensional vector x of non-negative integers and a move from configuration x to configuration y is possible iff x-y belongs to D. Starting from a game configuration, the players alternate their moves and the player that is unable to make a move loses the game. Because of the positivity condition, a match of the game always terminates after O(M^d), where M is the infinity norm of the starting d-dimensional vector.

The class of FDG games generalizes the game of Nim together with several of its variants. More precisely, while in the game of Nim and its variants a player can only remove pebbles from the heaps, in FDG a player can even add pebbles to the heaps but the overall number of pebbles strictly decreases after each move due to the positivity condition satisfied by each vector in D.

The authors of this paper generalize the arguments used by Larsson and Wästlund (E-JC 2013) and prove the existence of an FDG that is EXP-complete and can be solved in 2^{\Omega(n)} time - up to polynomial speed up - where n is the size of the input. From the paper itself, it is not clear which are the new ideas that, combined with the construction of Larsson and Wästlund, allow the authors to prove their main result (maybe the authors can be more explicit here).

The paper is concluded with a nice section containing a couple of interesting conjectures – the second of which would already imply the first one – which relates the problem of computing the P-positions to the Presburger arithmetic.

The paper is nicely written, easy to follow, and, in my opinion, it provides an interesting contribution.

ADDITIONAL COMMENTS:

  • Line 108, proof: KAYELS -> KAYLES
  • Line 131: Posiible -> Possible
  • Line 148-149: specify what Å means

Author Response

Line 108, proof: KAYELS -> KAYLES
Line 131: Posiible -> Possible
Response: fixed

Line 148-149: specify what Å means
Response: we are terribly sorry, but we do not understand the comment. There is no Å in these lines. But we add the name of the transition function with the whole list of its arguments: $x_{-1}$, $x_0$, $x_1$.